# OptiBreech collaborative care versus standard care for women with a breech-presenting fetus at term: A pilot parallel group randomised trial to evaluate the feasibility of a randomised trial nested within a cohort

**Shawn Walker** [1,2¤]*, **Emma Spillane** [3], **Kate Stringer**[4], **Lauren Trepte**[2], **Siân M. Davies**[1], **Jacana Bresson**[1], **Jane Sandall**[1], **Andrew Shennan**[1], **the OptiBreech Collaborative**[¶]

1 Faculty of Life Sciences & Medicine, Department of Women & Children's Health, School of Life Course & Population Sciences, King's College London, London, United Kingdom, 2 Women's and Children's Services, Chelsea and Westminster Hospital NHS Foundation Trust, London, United Kingdom, 3 Kingston Maternity, Kingston Hospital NHS Foundation Trust, Kingston upon Thames, Surrey, United Kingdom, 4 Women's Services, Surrey and Sussex Healthcare NHS Trust, East Surrey Hospital, Redhill, United Kingdom

¤ Current address: Faculty of Medicine, Imperial College London, London, United Kingdom
¶ Membership of the OptiBreech Collaborative is provided in the Acknowledgments
* Shawn.Walker1@nhs.net

**Data Availability Statement:** The data underlying the results presented in the study are available

## Abstract

OptiBreech collaborative care is a multi-disciplinary care pathway for breech presentation at term, with continuity from a breech specialist midwife, including where chosen, for vaginal breech birth (VBB). Pilot randomised trial using unblinded 1:1 parallel group allocation to OptiBreech versus standard care, within a cohort. Participants were women with a breech-presenting fetus > 33 weeks, at four sites in England, January–June 2022. A two-stage consent process was used. Participants consented to undergo random selection to be offered a 'new care process', with a choice to accept it, or not. Primary objectives were to identify recruitment, acceptance, and attrition rates. Randomisation procedures and potential primary outcomes for a substantive study were also feasibility-tested. 68 women were randomised between January–June 2022. The consent process was acceptable to participants, but randomisation was unacceptable to women who specifically sought OptiBreech care. Two women withdrew due to concerns about sharing personal information. More women planned a VBB when randomised to OptiBreech Care (23.5% vs 0, p = .002, 95% CI = 9.3%,37.8%). Women randomised to OptiBreech care had: lower rates of cephalic presentation at birth (38.2% vs 54.5%), higher rates of vaginal birth (32.4% vs 24.2%), lower rates of in-labour caesarean birth (20.6% vs 36.4%), lower rates of neonatal intensive care (5.9% vs 9.1%), and lower rates of severe neonatal morbidity (2.9% vs 9.1%). Randomisation was stopped on the advice of the steering committee before the planned sample of 104, as lack of access to VBB within standard care prohibited comparison of outcomes. Demand for VBB is sufficient for a cohort study, but comparison of outcomes by 1:1 randomisation is not feasible. OptiBreech care would be best evaluated using stepped wedge cluster

from: https://figshare.com/collections/OptiBreech_Care_IRAS_303028/6386370.

**Funding:** SW is funded by a National Institute of Health and Care Research (NIHR) Advanced Fellowship (300582; https://fundingawards.nihr.ac.uk/award/NIHR300582). JS at King's College London is an NIHR Senior Investigator. Both are supported by the NIHR Applied Research Collaboration South London (NIHR ARC South London) at King's College Hospital NHS Foundation Trust. The views expressed are those of the author(s) and not necessarily those of the NIHR or the Department of Health and Social Care, who played no role in study design, data collection and analysis, decision to publish, or preparation of the manuscript. Breech Birth Network provided training, educational resources, and funding for conference presentations.

**Competing interests:** I have read the journal's policy and the authors of this manuscript have the following competing interests: SW and ES are co-Directors of Breech Birth Network, Community Interest Company, a not-for-profit social enterprise that delivers VBB training and supports research. They and other members of the OptiBreech Collaborative have received teaching fees and expenses for providing breech training. This does not alter our adherence to PLOS ONE policies on sharing data and materials.

randomisation. Funded by the United Kingdom National Institute for Health and Care Research (NIHR300582). Clinical trial registration: ISRCTN 14521381.

## Introduction

OptiBreech collaborative care is a specialist, multi-disciplinary pathway for women with a breech-presenting fetus at term, developed out of previous research and in collaboration with service users [1–4]. OptiBreech care includes continuity from a breech specialist midwife and intrapartum care from professionals who have completed advanced training in physiological breech birth [5,6], wherever possible. Current United Kingdom (UK) National Health Service (NHS) recommendations for management of breech presentation at term are to offer external cephalic version (ECV) to turn the baby head-down, and if this fails to offer a pre-labour caesarean birth (CB) [7]. Although guidelines also support the choice of vaginal breech birth (VBB) [7–9], women find it difficult to access support for a planned VBB within current standard care, and this is in part due to very low overall clinical experience levels [10,11]. Little is known about the potential demand for planned VBB within a multi-disciplinary collaborative care model with skilled support in labour and what the outcomes might be if this were introduced nationally.

The aim of this pilot trial was to determine the feasibility of conducting a randomised trial nested within a cohort study comparing OptiBreech collaborative care with standard care for women with a breech pregnancy at term. Patient and Public Involvement and Engagement (PPIE) work aimed to ensure the study's design and interpretation were influenced by women who have lived experience of planning or attempting to plan a VBB because previous research indicated this population was least well served within current NHS standard care [1,10–12] (S1 Checklist).

## Materials and methods

### Study design

This pilot trial used a randomised, parallel group design with 1:1 allocation, nested within an observational cohort study. Women in the observational cohort had all requested OptiBreech care. This design was chosen for two reasons. Firstly, it enabled identification of preliminary safety outcomes among a larger cohort of women planning a VBB with OptiBreech care, some of whom may not be eligible for or may not consent to randomisation. Secondly, we anticipated that the larger cohort may host multiple nested randomised trials in the future, to increase the efficiency of delivering trials within this population [13].

The setting was four NHS Hospitals in England. Sites were chosen based on their recruitment rate and fidelity to protocol performance in the OptiBreech 1 observational study of planned VBBs [1,14].

The study was funded by the UK National Institute for Health and Care Research (NIHR, 300582) and sponsored by King's College London. Ethics approval was obtained from the West London & GTAC Research Ethics Committee (21/LO/0808, 19 November 2021). The pilot trial was prospectively registered with the ISRCTN (14521381, 18 October 2021).

### PPIE

Stakeholder involvement was facilitated through multiple public meetings, held in person and on-line during the research design stage [15]. Two members of the Trial Steering Committee

and one member of the research team were service users with lived experience of planning a VBB. Additional service users with lived experience of planning a VBB participated as members of the research team during qualitative work to refine the OptiBreech care pathway intervention [1,11] and consensus work to identify and prioritise outcome measures [16,17].

Service users influenced the design of the trial in multiple ways. They influenced the decision to begin the care pathway prior to 36 weeks. They emphasised the importance of providing earlier information and time to make decisions about care that would take place at 36 weeks. They carefully reviewed the consent process. Service user representatives advocated that all women who were randomised should still retain the option to accept or decline ECV, VBB or CB. This resulted in a pragmatic trial design suited to evaluating overall effectiveness of a care pathway, rather than the efficacy of one specific intervention over another. This also influenced the inclusion of the non-randomised cohort of women who were specifically requesting OptiBreech care. Service users were very keen that any results would not be used to remove choice and control, including the choice of planning a caesarean birth for women who preferred that.

Finally, PPIE input influenced the sub-category analyses. Our PPIE group prioritised knowing how outcomes for VBB compared with those for cephalic birth, and this was included as a planned sub-group analysis. Later, service users advocated for an equity analysis due to growing awareness of increased risk of adverse outcomes among women of minoritised ethnicity and skin colour in the UK [18]. Demographic information has been reported in line with the latest NICE style guide [19].

## Participants

Randomised participants were recruited between 10 January 2022 and 09 June 2022. Cohort participants included in this report were recruited until 28 June 2022 (Fig 1). The trial planned to recruit 104 women, but randomisation was stopped after 68 women were randomised. Inclusion and exclusion criteria for the cohort and nested randomised trial are listed in Table 1. Participants who were ineligible for or declined randomisation due to a request for the OptiBreech care pathway were recruited to the observational cohort arm only.

Consent process. Participants were recruited following a referral for counselling and/or care relating to breech presentation in the third trimester. An innovative and person-centred two-stage consent process was used, in which participants are only provided with information that is relevant to them [20,21]. All participants received information about the cohort study, including what data would be collected and what would be required from them to participate. This also explained randomisation, but not the specific intervention (a collaborative care pathway) being tested. Potential participants were instead informed that they would be selected by chance to be offered a 'new care process,' which they could accept or decline. Participants randomised to OptiBreech care were offered collaborative care led by a specialist midwife. They were informed that this was a new, untested care pathway and that they could request standard obstetric care if they preferred. Participants randomised to standard care were given the RCOG *Breech baby at the end of pregnancy patient information leaflet* [22] and scheduled for further counselling with their named obstetric consultant and an attempt at ECV if accepted.

Each participant provided consent to participate via written or e-consent form. Demographic data on ethnicity and gender was self-reported at the same time.

## Randomisation and masking

Eligible participants from the cohort were randomised to either standard care or OptiBreech care. The randomisation schedule was computer-generated using a dynamic allocation process

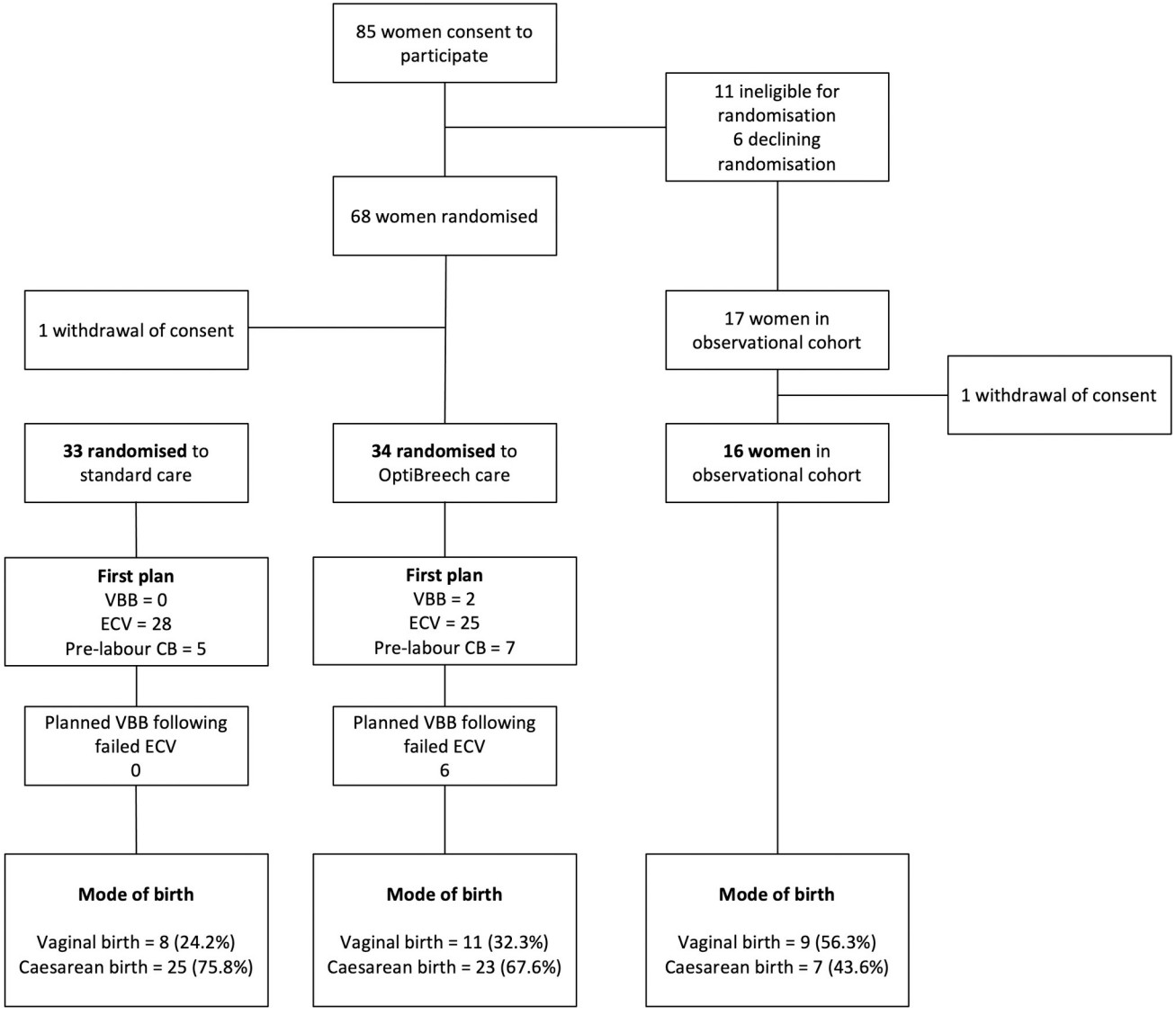

**Fig 1. OptiBreech care trial profile.**

through MedSciNet software. Minimisation factors included site, previous vaginal births (0 vs 1 or more), type of breech presentation (extended/frank vs any other or unknown), and gestation at enrolment (<36 weeks, 36–38+6, 39+ weeks). Allocation was scheduled to be equal.

Allocation occurred during the enrolment process within the MedSciNet e-Case Report Form (e-CRF) database and revealed to the person taking consent. Consent and randomisation were completed either by a Clinical Research Network midwife, the Principal Investigator (PI) or the Breech Lead Midwife/Obstetrician. Person-identifiable information, once entered into the database, was automatically moved onto a separate database. The database containing person-identifiable information was accessible only through a separate login and password, to protect participants' confidentiality. Due to the nature of the care pathway intervention, it was not possible to blind participants nor maternity care professionals. The neonatal teams assessing neonatal outcomes were not informed whether women were enrolled on the trial. Analysis of data was not blinded at this feasibility stage.

**Table 1. Inclusion and exclusion criteria for the cohort and nested randomised trial.**

| |
|---|
| Inclusion criteria for the OptiBreech cohort are: |
| •Live, singleton pregnancy with a breech-presenting fetus confirmed by ultrasound scan; |
| • Over 16 years of age; |
| •Referred for specialist care for breech presentation antenatally from 32 weeks; |
| •Breech presentation from 37 weeks discovered in labour; |
| •Requesting or preferring a vaginal birth; and |
| •Giving informed consent to participate to contribute data to the cohort study. |
| Exclusion criteria for the cohort are: |
| •Absolute reason for caesarean section already exists (eg. placenta praevia major); |
| •Requesting a caesarean section prior to recruitment; |
| •Multiple pregnancy; |
| •Life-threatening congenital anomaly; or |
| •Not consenting to contribute data to the cohort study |
| In addition to the cohort inclusion/exclusion criteria, eligibility for the randomised trial were: |
| •Consent to randomisation. |
| Exclusion criteria for randomisation were: |
| •Has already had an ECV attempt prior to recruitment; |
| •Rhesus isoimmunisation; |
| •Current or recent (less than 1 week) vaginal bleeding; |
| •Evidence of antenatal fetal compromise, including abnormal electronic fetal monitoring; |
| •Rupture of the membranes; |
| •Hyperextended neck on ultrasound; |
| •Estimated fetal weight less than 2000 g or less than $10^{th}$ centile at recruitment (if a growth scan has been performed); |
| •Estimated fetal weight greater than 3800g or over $95^{th}$ centile at recruitment (if a growth scan has been performed); |
| •Standing / footling presentation at the time of recruitment, defined as hips extended and breech above the inlet to the pelvis or not longitudinal; |
| •Any indication at the time of recruitment for induction to be recommended prior to 41 weeks of pregnancy, e.g. gestational diabetes, obstetric cholestasis, advanced maternal age; |
| •Breech diagnosed for the first time in labour; and |
| •2 or more previous caesarean sections; |
| •and any other exclusion criteria for either ECV or vaginal breech birth in the current RCOG guidelines. |

## Procedures

Participants in the standard care arm were offered ECV as a first-line intervention and/or referred to their named obstetric consultant's antenatal clinic for further counselling regarding mode of birth if declined. Participants in the OptiBreech care arm were counselled by a member of the OptiBreech team and were offered the option of planning a VBB with OptiBreech support, attempting an ECV or planning a pre-labour CB from 39 weeks gestation. All Opti-Breech care was co-ordinated by a breech specialist midwife. A full TIDieR checklist available within the prospectively registered protocol [23,24].

## Outcomes

This report concerns the trial's short-term feasibility and safety outcomes. These are listed in Table 2, along with any variations from the original protocol in the way they are reported. Adverse events related to study procedures were assessed via reports by PIs included in the local site files. The criteria to assess trial feasibility were set in agreement with the Trial Steering Committee prior to the start of randomisation and outlined in the protocol [24].

**Table 2. Outcome measures.**

| *Outcome in protocol* | *Outcome reported* |
|---|---|
| **Primary outcomes** | |
| Recruitment rate recorded as the number of eligible participants who consent to participate in the study by 6 months (randomised) and overall (non-randomised). | Reported as planned. Because site opening times were unpredictable during the COVID-19 pandemic and randomisation was stopped early on the advice of the TSC, monthly mean recruitment figures according to the number of months each site was open are also reported, to enable more accurate estimations for future trials. |
| Acceptance rate recorded as the number of participants randomised to OptiBreech Care who plan a vaginal breech birth, and the number of participants randomised to the control who attempt an ECV, measured at the time of birth. | Reported as planned. It was observed that acceptance rate was only one factor that influenced the flow of care through the standard and OptiBreech pathways. These additional factors would need to be taken into consideration to measure the full impact of the pathways. Therefore, the following outcomes were also reported: 1. Planned VBB following initial counselling and at any point, eg. including following a failed ECV or upon arrival in labour with an unexpected breech presentation; 2. Requesting ECV following initial counselling; 3. Total attempts planned, including second attempts; 4. Number of ECVs attempted for eligible women requesting an ECV; 5. Number of attempted ECVs successful; 6. Number of ECVs not performed as planned, for reasons other than spontaneous cephalic version; 7. Reasons ECV not performed as planned; 8. Planned pre-labour caesarean birth following initial counselling. |
| Attrition rate recorded as the number of participants who consent to participate who remain in the study until the end of follow-up at 4 months after birth; and Long-term attrition rate recorded as the number of OptiBreech 1 participants who complete 1-year and 2-year follow-up surveys when invited. | In this report, attrition is reported as the number of participants who consented to participate and subsequently withdrew consent to contribute their data. Results of longer-term follow-up outcomes and survey completion rates will be reported separately. |
| Fidelity to intervention recorded as number of planned VBBs attended by a proficient team member, measured at the time of birth. Proficiency defined in protocol as: A professional is considered currently proficient to facilitate OptiBreech care if they have: 1) Participated in 6 hours of evaluated physiological breech birth training;[5] 2) Attended at least 10 vaginal breech births, including resolution of complications using manual manoeuvres; 3) Attended or taught in simulation at least 3 vaginal breech births within the past year; 4) Delivered physiological breech birth training at least once within the past year, including reflective reviews of births attended; 5) Completed an OptiBreech Proficiency self-assessment and indicated that they feel competent to implement the OptiBreech Practice Guideline at vaginal breech births where they are the designated clinical lead, and this has been confirmed by the OptiBreech Leads. | Reported as planned. Also reported as number of planned VBBs attended by trained members of the team, consistent with previous reports. |
| Costs to deliver the service recorded as total number of days and nights spent on call to support planned VBBs in the trial by 6 months. | Costs will be reported in a separate economic report. |
| **Secondary outcomes** | |

(*Continued*)

**Table 2.**  (Continued)

| Outcome in protocol | Outcome reported |
|---|---|
| **Primary outcomes** | |
| Admission to higher-level neonatal care, measured at 28 days following birth, as a binary (yes/no) and continuous (number of days/nights) outcome, from patients' medical records. | Reported as planned. |
| Mode of birth measured using patient's medical records on day of birth, as a categorial measurement to include the following categories: vaginal breech birth, forceps breech, pre-labour CS, emergency CS, cephalic vaginal birth, cephalic forceps, cephalic ventouse. | Reported as planned. Exception: in line with new guidance published about the use of language preferred by service users, we refer to pre-labour and emergency caesarean sections as 'pre-labour caesarean birth' and 'in-labour caesarean birth' [25]In-labour caesarean birth included:• Category 1. Immediate threat to the life of the woman or fetus (for example, suspected uterine rupture, major placental abruption, cord prolapse, fetal hypoxia or persistent fetal bradycardia).• Category 2. Maternal or fetal compromise which is not immediate life-threatening.Pre-labour caesarean birth included:• Category 3. No maternal or fetal compromise but needs early birth.• Category 4. Birth timed to suit woman or healthcare provider. |
| Composite neonatal perinatal death or serious adverse morbidity, measured at 28 days following birth, from patients' medical notes; serious neonatal morbidity to include the following: 5 minute APGAR score <7, peripheral nerve injury present at discharge from hospital, skull fracture, spinal cord injury, admission to NICU>4 days, intubation/ventilation >24 hours, convulsions >24 hours, parenteral or tube feeding >24 hours. | Reported as planned. Admission to both SCBU and NICU included in admission rate, as not all sites had a NICU. |
| Composite maternal death or serious morbidity, measured at 28 days following birth, from patients' medical notes; serious maternal morbidity to include the following: postpartum haemorrhage >1000 mL, obstetric anal sphincter injury, cervical laceration involving lower uterine segment, vertical uterine incision or serious extension to transverse uterine incision, bladder, ureter or bowel injury requiring repair, dilation and curettage for bleeding or retained placental tissue, manual removal of placenta, uterine rupture, hysterectomy, vulval or perineal haematoma requiring evacuation, wound dehiscence / breakdown, wound infection requiring prolonged hospital stay / readmission / antibiotics, sepsis, disseminated intravascular coagulation. | Reported as planned. Exception: PPH >1000 mL was an error in the protocol. This should have been >1500 mL for severe morbidity, to harmonise with our previous reports. In this report, only cases of PPH > 1500 mL are included as a measure of severe morbidity. |
| Use of services following referral for breech care, to include antenatal and postnatal appointments, total time spent admitted to hospital, number of ECVs, number of ultrasound scans, and professionals present at birth, measured at 28 days following birth from patients' medical notes. | Costs will be reported in a separate economic report. |
| Satisfaction with care, measured using previously validated survey questions with a 5-point Likert scale, at 1 month post birth. | Follow-up surveys and longer-term outcomes will be reported separately. |
| Experience of childbirth, measured using the 'Childbirth Experience Questionnaire' [26,27] at 1 month post birth. | Follow-up surveys and longer-term outcomes will be reported separately. |
| Health-related quality of life, using the PROMIS-10 survey [28] at 1 month, 3–4 months, 1 year and 2 years following birth. | Follow-up surveys and longer-term outcomes will be reported separately. |

(*Continued*)

**Table 2.** (Continued)

| Outcome in protocol | Outcome reported |
|---|---|
| **Primary outcomes** | |
| Infant's development, using the appropriate Ages and Stages Questionnaires at 3–4 months, 1 year and 2 years following birth [29]. | Follow-up surveys and longer-term outcomes will be reported separately. |

**Abbreviations:** TSC = trial steering committee; ECV = external cephalic version; CS = caesarean birth; NICU = neonatal intensive care unit; SCBU = special care baby unit; PPH = post-partum haemorrhage.

## Statistical analysis

The sample size of 104 women was calculated to enable estimation of a recruitment rate between 20–80%, with a 95% confidence interval, within ±10%. For a site with an average of 25 women it would be possible to estimate the same recruitment rate within ±18%.

Primary outcomes were assessed by intention to treat according to randomisation arm. Participants who withdrew their data were excluded from analysis. For recruitment rates, mean and confidence intervals were calculated using a one-sample t-test. Site opening times were very unpredictable during the COVID-19 pandemic, and one site closed early. Therefore, recruitment figures were calculated according to the number of months each site was open. Secondary outcomes, safety and feasibility outcomes were assessed by intention to treat (randomisation arm and cohort).

The OptiBreech care intervention was designed to enable women to choose the option of VBB, which women reported as being unavailable or inaccessible under standard care. Acceptance was measured as the rate of planning a VBB. This was measured following initial counselling and 'at any point,' including after failed/non-attempted ECV. The significance of the difference in percentage of participants planning a VBB was calculated using a Fisher exact test. The 95% confidence intervals were calculated using a two-sided z-test (Wald and with Agresti-Caffo adjustments). Although the feasibility pilot was not designed and not powered to detect significant differences in outcomes, the significance of this finding is reported because it was the basis of the Trial Steering Committee (TSC) decision to stop randomisation early, during an unplanned interim analysis.

Planned sub-group analysis was performed for the following groups, from the entire cohort: 1) those who planned a VBB at any point versus those who had not; 2) actual VBBs versus vaginal cephalic births; and 3) presentation on admission for labour/birth care. These subgroup analyses were chosen because 1) this is the comparison most often used in large observational cohort studies; 2) this comparison was prioritised by PPI group members; and 3) cephalic presentation at birth is evaluated as an outcome in Cochrane Reviews concerning the management of breech presentation at term. An unplanned sub-group analysis was performed for minoritised ethnicity (non-British and Black or Brown/mixed cohorts), as advocated by the PPI group.

Data were analysed using SPSS Version 28. For the feasibility study, because of the small sample size and non-clinical primary outcomes, the ethics review approved the Trial Steering Committee to serve as the Data Monitoring Committee.

## Role of the funding source

The National Institute for Health and Care Research (NIHR) had no role in study design, data collection, data analysis, data interpretation, or writing of the report. The unplanned interim analysis was prompted by the Department of Health and Social Care's Research Reset

**Table 3. Recruitment rates.**

| Code | Months open | Randomised Participants | Monthly mean | Cohort participants | Monthly mean | Total |
|------|-------------|-------------------------|--------------|---------------------|--------------|-------|
| 100-A | 2 | 14 | 7.00 | 0 | .00 | 14 |
| 100-B | 5 | 25 | 5.00 | 11 | 2.20 | 36 |
| 101 | 5 | 20 | 4.00 | 1 | .20 | 21 |
| 102 | 4 | 8 | 2.00 | 4 | 1.00 | 12 |
| **Total** | 16 Mean: 4.00 St Dev: 1.41 | 67 Mean: 16.75 St Dev: 7.37 | 4.50 (95% CI) (1.19–7.81) | 16 Mean: 4.00 St Dev: 4.97 | .85 (95% CI) (-.74–2.44) | 83 Mean: 20.75 St Dev 10.87 |

programme, which began in March 2022 to clear a backlog of open clinical research studies following the COVID-19 pandemic [30] As part of this process, the sponsor requested a review, and the TSC were asked to be involved in this.

Breech Birth Network, a not-for-profit Community Interest Company, provided training for OptiBreech teams, educational resources, and funding for conference presentations.

## Results

### Recruitment rates

Recruitment rates are presented in Table 3.

### Demographics

Basic demographics of the sample are presented in Table 4. A complete data set was obtained, excepting one BMI measurement. Minimisation factors resulted in comparable randomised samples based on mean gestation at enrolment, parity and type of breech presentation. The cohort arm was characterised a higher mean gestation at enrolment, more extended breech presentations and more multiparous women.

### Acceptance

All women randomised to OptiBreech collaborative care accepted that care pathway, including care co-ordinated by a breech specialist midwife (100% acceptance). Patterns of service usage between the two models of care are reported in Table 5. Compared to standard care, more women planned a VBB at some point within the OptiBreech care pathway (0/33 vs 8/34, 23.5%). This difference was statistically significant, z = -2.969, p(2-tailed) = 0.003, Wald 95% CI [9.3%,37.8%], Agresti-Caffo corrected 95% CI [7.0%,37.3%]. Although more women initially accepted an attempt at ECV within standard care, more actual attempts occurred as planned within the OptiBreech care pathway.

### Attrition

Two women withdrew from the study. One was randomised and the other had consented to participate in the cohort study. Both were planning a VBB. Their concerns related to sharing personal information about themselves and their babies, and their data were not included in the analyses. A total of 21/67 (31.3%) women declined an ECV attempt, including those who changed their minds after initially accepting the offer.

### Fidelity

During the pilot trial, 3/7 (43%) of the VBBs were attended by an OptiBreech team member meeting all proficiency criteria [1] The research team investigated all instances where this

**Table 4. Baseline characteristics by intention-to-treat.**

| Outcome | Randomised Standard Care | Randomised OptiBreech Care | Cohort OptiBreech Care | Total |
|---|---|---|---|---|
| N = (%) | 33 | 34 | 16 | 83 |
| **Gestation** | | | | |
| Mean at enrolment (weeks+days) | 35+6 | 35+5 | 38+2 | 36+2 |
| Mean at birth (weeks+days) | 39+5 | 39+2 | 40+0 | 39+4 |
| Gestational week at birth | | | | |
| 36 | 1 (3.0%) | 2 (5.9%) | 0 | 3 (3.6%) |
| 37 | 4 (12.1%) | 1 (2.9%) | 0 | 5 (6.0%) |
| 38 | 3 (9.1%) | 6 (17.6%) | 2 (12.5%) | 11 (13.3%) |
| 39 | 13 (39.4%) | 18 (52.9%) | 6 (37.5%) | 37 (44.6%) |
| 40 | 5 (15.2%) | 4 (11.8%) | 4 (25.0%) | 13 (15.7%) |
| 41 | 4 (12.1%) | 3 (8.8%) | 3 (18.8%) | 10 (12.0%) |
| 42 | 3 (9.1%) | 0 | 1 (6.3%) | 4 (4.8%) |
| **Source of referral** | | | | |
| midwife | 12 (36.4%) | 10 (29.4%) | 8 (50.0%) | 30 (36.1%) |
| obstetrician | 3 (9.1%) | 2 (5.9%) | 1 (6.3%) | 6 (7.2%) |
| sonographer | 18 (54.5%) | 22 (64.7%) | 2 (12.5%) | 42 (50.6%) |
| self (originally booked elsewhere) | 0 | 0 | 5 (31.3%) | 5 (6.0%) |
| **Previous vaginal births** | | | | |
| none | 24 (72.7%) | 24 (70.6%) | 9 (56.3%) | 57 (69.7%) |
| one or more | 9 (27.3%) | 10 (29.4%) | 7 (43.8%) | 26 (31.3%) |
| Parity | | | | |
| 0 | 21 (63.6%) | 22 (64.7%) | 9 (56.3%) | 52 (62.7%) |
| 1 | 8 (24.2%) | 8 (23.5%) | 5 (31.3%) | 21 (25.3%) |
| 2 | 3 (9.1%) | 4 (11.8%) | 2 (12.5%) | 9 (10.8%) |
| 3 | 1 (3.0%) | 0 | 0 | 1 (1.2%) |
| **Type of breech presentation** | | | | |
| extended / frank | 21 (63.6%) | 20 (58.8%) | 12 (75.0%) | 53 (63.9%) |
| any other or uncertain | 12 (36.4%) | 14 (41.2%) | 4 (25.0%) | 30 (36.1%) |
| **Maternal demographics** | | | | |
| Age at booking (mean years, std dev) | 32.4 (5.85) | 32.3 (5.84) | 31.7 (6.19) | 32.2 (5.85) |
| BMI (mean, std dev) | 23.8 (4.20) | 25.5 (5.57) | 23.167 (4.37) | 24.4 (4.88) |
| **Self-reported variables** | | | | |
| Gender | | | | |
| female | 33 (100%) | 34 (100%) | 16 (100%) | 83 (100%) |
| male (trans) | 0 | 0 | 0 | 0 |
| non-binary | 0 | 0 | 0 | 0 |
| Ethnic group | | | | |
| Scottish / English / Welsh / Northern Irish / British | 16 (48.5%) | 14 (41.2%) | 4 (25.0%) | 34 (41.0%) |
| Irish | 0 | 1 (2.9%) | 0 | 1 (1.2%) |
| Any other white background | 9 (27.3%) | 12 (35.3%) | 3 (18.8%) | 24 (28.9%) |
| White and Black Caribbean | 0 | 1 (2.9%) | 0 | 1 (1.2%) |
| White and Black African | 1 (3.0%) | 0 | 1 (6.3%) | 2 (2.4%) |
| Indian | 2 (6.1%) | 1 (2.9%) | 3 (18.8%) | 6 (7.2%) |
| Pakistani | 1 (3.0%) | 0 | 2 (12.5%) | 3 (3.6%) |
| Bangladeshi | 1 (3.0%) | 1 (2.9%) | 0 | 2 (2.4%) |
| Any other Asian background | 1 (3.0%) | 3 (8.8%) | 3 (18.8%) | 7 (8.4%) |

*(Continued)*

**Table 4.** (Continued)

| Outcome | Randomised Standard Care | Randomised OptiBreech Care | Cohort OptiBreech Care | Total |
|---|---|---|---|---|
| African | 0 | 1 (2.9%) | 0 | 1 (1.2%) |
| Arab | 2 (6.1%) | 0 | 0 | 2 (2.4%) |
| Non-British | 17 (51.5%) | 20 (58.8%) | 12 (75.0%) | 49 (59.0%) |
| Non-white | 8 (24.2%) | 7 (20.6%) | 9 (56.3%) | 24 (28.9%) |
| Interpreter required | 1 (3.0%) | 4 (11.8%) | 2 (12.5%) | 7 (8.4%) |

criterion was not met. Of the four, one was attended by someone who had completed training but did not meet the proficiency criteria (4/7, 57%). The other three, including one breech presentation diagnosed late in labour, progressed quickly and completed before the OptiBreech team member arrived.

## Safety and mode of birth outcomes

Analysis by intention to treat is presented in Table 6. A complete data set was obtained. Compared to women randomised to standard care, women randomised to OptiBreech care had: lower rates of cephalic presentation at birth (38.2% vs 54.5%), higher rates of vaginal birth (32.4% vs 24.2%), lower rates of in-labour caesarean birth (20.6% vs 36.4%), lower rates of neonatal intensive care (5.9% vs 9.1%), and lower rates of severe neonatal morbidity (2.9% vs 9.1%).

## Sub-group analyses

Planned sub-group analyses are presented in Table 7. Safety was assessed as planned using the secondary outcomes. Within the entire cohort, breech presentation on admission to

**Table 5.** Acceptance / Treatment plans.

| | Randomised standard care | Randomised OptiBreech care |
|---|---|---|
| Acceptance criteria | n = 33 | n = 34 |
| Planning a VBB | | |
| Following initial counselling | 0 | 2 (5.9%) |
| At any point | 0 | 8 (23.5%) |
| Requesting attempt at ECV | | |
| Planned following initial counselling | 28 (84.8%) | 25 (73.5%) |
| Total attempts planned, including 2nd attempts | 32 | 31 |
| Spontaneous cephalic version prior to attempt | 8 (24.2%) | 3 (8.8%) |
| Eligible (breech) for attempt as planned | 24 | 28 |
| Number of ECVs attempted | 14/24 (58.3%) | 23/28 (82.1%) |
| Number of attempted ECVs successful | 8/14 (57.1%) | 11/23 (47.8%) |
| Number of planned ECVs not performed as planned, for reasons other than spontaneous version | 10/24 (41.6%) | 5/28 (17.9%) |
| Reasons ECVs not performed as planned | Labour ward activity (4) Changed mind / declined (4) Clinical advice (1) Labour before ECV (1) | Labour ward activity (1) Changed mind (3) Clinical advice (1) |
| Requesting pre-labour caesarean birth | | |
| Following initial counselling (no ECV) | 5 (15.2%) | 7 (20.6%) |

**Table 6. Analysis by intention to treat.**

| Outcome | Randomised Standard Care | Randomised OptiBreech Care | Cohort OptiBreech Care | Total |
|---|---|---|---|---|
| N (%), SD = standard deviation | 33 | 34 | 16 | 83 |
| **Presentation on admission for labour/birth** | | | | |
| breech | 15 (45.5%) | 20 (58.8%) | 9 (56.3%) | 44 (53.0%) |
| cephalic | 18 (54.5%) | 13 (38.2%) | 7 (43.8%) | 38 (45.8%) |
| transverse | 0 | 1 (2.9%) | 0 | 1 (1.2%) |
| **Mode of Birth** | | | | |
| vaginal breech birth | 0 | 1 (2.9%) | 5 (31%) | 6 (7.2%) |
| forceps breech birth | 0 | 0 | 1 (6.3%) | 1 (1.2%) |
| cephalic vaginal birth | 8 (24.2%) | 7 (20.6%) | 3 (18.8%) | 18 (21.7%) |
| cephalic ventouse birth | 0 | 3 (8.8%) | 0 | 3 (3.6%) |
| cephalic forceps birth | 0 | 0 | 0 | 0 |
| in-labour caesarean birth (Cat 1/2) | 12 (36.4%) | 7 (20.6%) | 5 (31.3%) | 24 (28.9%) |
| pre-labour caesarean birth (Cat 3/4) | 13 (39.4%) | 16 (47.1%) | 2 (12.5%) | 31 (37.3%) |
| TOTAL vaginal birth | 8 (24.2%) | 11 (32.3%) | 9 (56.3%) | 28 (33.7%) |
| TOTAL caesarean birth | 25 (75.8%) | 23 (67.6%) | 7 (43.6%) | 55 (66.3%) |
| **Higher-level care** | | | | |
| Admission to NICU or SCBU | 3 (9.1%) | 2 (5.9%) | 0 | 5 (6.0%) |
| Mean NICU/SCBU nights | .33 *(SD 1.24)* | .24 *(SD 1.21)* | 0 | .23 *(SD 1.10)* |
| Maternal admission to HDU | 2 (6.1%) | 2 (5.9%) | 1 (6.3%) | 5 (6.0%) |
| Mean HDU nights | .06 *(SD .24)* | .06 *(SD .24)* | .06 *(SD .25)* | .06 *(SD .24)* |
| Mean postnatal ward nights | 1.48 *(SD 1.06)* | 1.53 *(SD .96)* | 1.88 *(SD 1.41)* | 1.58 *(SD 1.10)* |
| **Adverse outcomes** | | | | |
| Apgar <7 at 5 minutes | 0 | 0 | 0 | 0 |
| Severe neonatal morbidity / mortality | 3 (9.1%) | 1 (2.9%) | 0 | 4 (4.8%) |
| Severe maternal morbidity / mortality | 5 (15.2%) | 4 (11.8%) | 1 (6.3%) | 10 (12.0%) |

labour/birth (n = 44), compared to cephalic presentation (n = 38), was associated with: lower levels of neonatal admission (2.3% versus 10.5%), lower levels of severe neonatal morbidity (2.3% vs 7.9%), fewer maternal admissions to HDU (4.5% vs 7.9%) and less severe maternal morbidity (13.6% vs 21.1%). There was no instance of Apgar <7 at 5 minutes or perinatal mortality following recruitment. Severe neonatal morbidity included: NICU admission >4 days (3), intubation/ventilation >24 hours (1), and parental or tube feeding >24 hours (1). Severe maternal morbidity included: EBL>1500 mL (3), OASI (1), vertical incision or serious extension to transverse uterine incision (2), manual removal of placenta (1), wound infection requiring prolonged hospital stay / readmission / antibiotics (1), and sepsis (2).

The sub-group analysis by ethnicity (Table 8) indicated that more non-British and Black or Brown women planned VBBs. This difference was most apparent within the randomised, OptiBreech care arm, where 30.0% of non-British participants and 28.6% of Black or Brown participants planned a VBB.

Adverse events related to study procedures were assessed via reports by PIs included in the local site files (Table 9). Four events were reported; each occurred in a separate site. Each was discussed at the Trial Steering Committee meeting, and the first three contributed to the decision to stop randomisation.

**Table 7. Subgroup analysis of entire cohort.**

| Outcome | Planned vaginal breech birth | No planned vaginal breech birth | Vaginal births | | Presentation on admission for labour/birth | | |
|---|---|---|---|---|---|---|---|
| | | | breech | cephalic | breech | cephalic | transverse |
| N = 83 (%) | 17 | 66 | 7 | 21 | 44 | 38 | 1 |
| **Presentation on admission for labour/birth** | | | | | | | |
| breech | 15 (88.2%) | 29 (43.9%) | 5 (71.4%) | 0 | | | |
| cephalic | 2 (11.8%) | 36 (54.5%) | 2* (28.5%) | 20 (95.2%) | | | |
| transverse | 0 | 1 (1.5%) | 0 | 1 (4.8%) | | | |
| **Mode of Birth** | | | | | | | |
| vaginal breech birth | 4 (23.5%) | 2 (3.0%) | 6 (85.7%) | | 4 (9.1%) | 2* (5.3%) | 0 |
| forceps breech birth | 1 (5.9%) | 0 | 1 (14.3%) | | 1 (2.3%) | 0 | 0 |
| cephalic vaginal birth | 1 (5.9%) | 17 (25.8%) | | 18 (85.7%) | 0 | 17 (44.7%) | 1 (100%) |
| cephalic ventouse birth | 0 | 3 (4.5%) | | 3 (14.3%) | 0 | 3 (7.9%) | 0 |
| cephalic forceps birth | 0 | 0 | | 0 | 0 | 0 | 0 |
| in-labour caesarean birth | 4 (23.5%) | 20 (30.3%) | | | 10 (22.7%) | 14 (36.8%) | 0 |
| pre-labour caesarean birth | 7 (41.2%) | 24 (36.4%) | | | 29 (65.9%) | 2 (5.3%) | 0 |
| TOTAL vaginal birth | 6 (35.3%) | 22 (33.3%) | | | 5 (11.4%) | 22 (57.9%) | 1 (100%) |
| TOTAL caesarean birth | 11 (64.7%) | 44 (66.7%) | | | 39 (88.6%) | 16 (42.1%) | 0 |
| **Higher-level care** | | | | | | | |
| Admission to NICU or SCBU | 1 (5.9%) | 4 (6.1%) | 0 | 3 (14.3%) | 1 (2.3%) | 4 (10.5%) | 0 |
| Mean NICU/SCBU nights | .06 (SD .24) | .27 (SD 1.22) | 0 | .57 (SD 1.72) | .14 (SD .91) | .34 (SD 1.300) | 0 |
| Maternal admission to HDU | 0 | 5 (7.6%) | 1 (14.3%) | 1 (4.8%) | 2 (4.5%) | 3 (7.9%) | 0 |
| Mean HDU nights | 0 | .08 (SD.27) | .14 (SD .38) | .05 (SD .22) | .05 (SD .21) | .08 (SD .27) | 0 |
| Mean postnatal ward nights | 1.76 (SD 1.09) | 1.53 (SD 1.10) | 1.71 (SD 1.50) | 1.05 (SD .97) | 1.73 (SD 1.09) | 1.45 (SD 1.08) | 0 |
| **Adverse outcomes** | | | | | | | |
| Apgar <7 at 5 minutes | 0 | 0 | 0 | 0 | 0 | 0 | 0 |
| Severe neonatal morbidity / mortality | 0 | 4 (6.1%) | 0 | 2 (9.5%) | 1 (2.3%) | 3 (7.9%) | 0 |
| Severe maternal morbidity / mortality | 3 (17.6%) | 11 (16.7%) | 1 (14.2%) | 3 (14.3%) | 6 (13.6%) | 8 (21.1%) | 0 |

## Substantive trial feasibility

The pilot trial met two pre-specified green and one amber substantive trial feasibility criteria. The minimum effectiveness target was met, as the vaginal birth rate among women randomised to OptiBreech care was >30% and greater than that among women randomised to standard care. The safety target was met, as there were no instances of Apgar <7 at 5 minutes or death in either arm. The trial fell into the 'amber' criteria for progression to a substantive trial for the recruitment target. Sites were on target to achieve the pre-specified sample; however, the original overall target was not met due to the decision to stop randomisation. Women only planned a VBB within the OptiBreech care cohort or when randomised to OptiBreech care. The TSC concluded that the demand for VBB was sufficient for a large observational cohort

**Table 8. Subgroup analysis of entire cohort by ethnicity.**

| Outcome | Ethnic group | | Skin colour | |
| --- | --- | --- | --- | --- |
| | **British** | **Non-British** | **White** | **Black or Brown** |
| N = (%) | 34 | 49 | 59 | 24 |
| **Planned vaginal breech birth** | | | | |
| yes | 5/34 (14.7%) | 12/49 (24.5%) | 11 (18.6%) | 6 (25.0%) |
| Randomised, standard care | 0/16 | 0/17 | 0/25 | 0/8 |
| Randomised, OptiBreech | 2/14 (14.3%) | 6/20 (30.0%) | 6/27 (22.2%) | 2/7 (28.6%) |
| OptiBreech cohort | 3/4 (75.0%) | 6/12 (50.0%) | 5/7 (71.4%) | 4/9 (44.4%) |
| no | 29/34 (85.3%) | 37/49 (75.5%) | 48 (81.4%) | 18 (75.0%) |
| **Mode of Birth** | | | | |
| vaginal breech birth | 2 (5.9%) | 4 (8.2%) | 2 (3.4%) | 4 (16.7%) |
| forceps breech birth | 1 (2.9%) | 0 | 1 (1.7%) | 0 |
| cephalic vaginal birth | 8 (23.5%) | 10 (20.4%) | 13 (22.0%) | 5 (20.8%) |
| cephalic ventouse birth | 2 (5.9%) | 1 (2.0%) | 3 (5.1%) | 0 |
| in-labour caesarean birth | 11 (32.4%) | 13 (26.5%) | 16 (27.1%) | 8 (33.3%) |
| pre-labour caesarean birth | 10 (29.4%) | 21 (42.9%) | 24 (40.7%) | 7 (29.2%) |
| TOTAL vaginal birth | 13 (38.2%) | 15 (30.6%) | 19 (32.2%) | 9 (37.5%) |
| TOTAL caesarean birth | 21 (61.8%) | 34 (69.4%) | 40 (67.8%) | 15 (62.5%) |
| **Adverse outcomes** | | | | |
| Severe neonatal morbidity / mortality | 4 (11.8%) | 0 | 4 (6.8%) | 0 |
| Severe maternal morbidity / mortality | 6 (17.6%) | 4 (8.2%) | 7 (11.9%) | 3 (12.5%) |

study, but 1:1 randomisation does not appear a feasible way of evaluating the safety of VBB between standard care and OptiBreech care. Because this was clear at the interim analysis, further randomisation was not perceived to add additional value.

## Discussion

The results of the OptiBreech care feasibility pilot trial indicate that women were willing to participate, teams were able to recruit, and data required to evaluate short-term safety outcomes were complete. However, 1:1 randomisation is not the optimal design to compare outcomes for VBBs planned within OptiBreech collaborative care to those within standard care.

Secondary outcomes were all positive, suggesting that OptiBreech care should be evaluated further, using a more appropriate trial design informed by this feasibility work. Care within

**Table 9. Adverse events reported in site files.**

1) **Medication error.** A medication error occurred in which a woman was given an incorrect dosage of terbutaline prior to an ECV attempt, 500 micrograms rather than 250 micrograms, within the standard care pathway.

2) **Protocol violation.** A protocol violation occurred in which the breech specialist midwife performed an ECV attempt for a woman who had been randomised to the standard care pathway. The woman's ECV had been cancelled and re-scheduled three times due to labour ward activity and unavailability of an obstetrician to perform the procedure. Following discussion with the local Principal Investigator (PI), it was decided that the woman's right to care should take precedence, and the ECV was provided within the OptiBreech care pathway.

3) **Refusal to provide collaborative care.** The lead site was paused to recruitment after the Labour Ward Lead reported that some consultant obstetricians had refused to be involved with births on the trial. An attempt was made to reinstate the site after discussion with the Lead Obstetrician. However, during the next actual planned VBB, the consultant obstetrician on-call recommended a caesarean birth because 'primip breech is not part of [his] practice.' He then suggested the birth should take place on the midwife-led unit rather than the obstetric unit, left the hospital and was not physically present to supervise the obstetric trainees when the VBB occurred. Although no adverse outcome resulted, this constituted a significant safety concern, and the site was permanently closed.

4) **Lack of research support.** One PI was informed that the research midwives would not be able to support studies 'led by midwives.' Following discussions with the research leadership within the Trust, this appeared to be a misunderstanding. A plan for providing support from the clinical research network midwives was put in place.

the OptiBreech pathway enabled greater access to all three guidelines-recommended care options, compared to standard care. When randomised to OptiBreech care, more eligible women had an ECV as planned and more planned a pre-labour CB as their first option. Compared so standard care, more women randomised to OptiBreech care planned a VBB (23.5% vs 0, p = .003, 95% CI = .093,.378).

Within this small feasibility sample, clinical outcomes were also better within the Opti-Breech care pathway compared to standard care. Despite having lower rates of cephalic presentation at birth (38.2% vs 54.5%), women randomised to OptiBreech care had: higher rates of vaginal birth (32.4% vs 24.2%), lower rates of in-labour caesarean birth (20.6% vs 36.4%), lower rates of neonatal intensive care (5.9% vs 9.1%), and lower rates of severe neonatal morbidity (2.9% vs 9.1%). Within the entire cohort, breech presentation on admission to labour/birth (n = 44), compared to cephalic presentation (n = 38), was associated with: lower levels of neonatal admission (2.3% versus 10.5%), lower levels of severe neonatal morbidity (2.3% vs 7.9%), fewer maternal admissions to HDU (4.5% vs 7.9%) and less severe maternal morbidity (13.6% vs 21.1%).

Recruitment to the cohort of women requesting OptiBreech collaborative care (.85/month, 95% CI .74–2.44) was in line with our previous implementation feasibility study of women planning a VBB (.90/month, 95% CI .64–1.16) [14] This suggests that, within an NHS model of care where the option of VBB is both accessible and acceptable, approximately one woman per month will choose to plan a VBB. Previous research has indicated that approximately one in three women elect to plan a VBB when offered balanced and supportive counselling [31,32]. This was consistent with the demand observed within the OptiBreech care pathway, yet no women planned a VBB when randomised to standard care. Multiple systematic reviews have suggested women encounter barriers to accessing support for a VBB within standard care, and that this option is only acceptable if the birth is likely to be attended by supportive and appropriately trained professionals [10,11]

This is also consistent with qualitative work involving women receiving OptiBreech care [1] and feedback from our PPIE group. In the qualitative study, women reported numerous examples of coercive counselling to dissuade them from attempting a VBB [1]. Participants reported that they received detailed and balanced counselling from breech specialist midwives and obstetricians, and this enabled them to access and plan the birth they preferred, including the option of pre-labour CB. In this pilot trial, marginally more women also chose to plan a pre-labour CB following initial counselling within the OptiBreech care pathway.

There is sufficient evidence to conclude that current standard care is unacceptable to women who wish to plan a VBB and that clinicians frequently lack sufficient training and/or equipoise [10,11]. The available evidence indicates that the OptiBreech care pathway is acceptable to women wishing to plan a VBB because it resolves many of the barriers inherent in standard care [1].

While we can say with confidence that OptiBreech collaborative care does significantly increase access to the choice of a VBB, we do not yet know how introducing this care pathway at scale will affect clinical outcomes. To date, we have evidence from three small studies, including this pilot trial, each of which indicate an improvement in outcomes compared to standard care [1,5]. We therefore propose that clinical outcomes should be evaluated using a stepped wedge cluster trial design. Roll-out within a stepped wedge cluster trial would immediately improve access to all guideline-recommended care options while simultaneously evaluating the safety outcomes to support future informed decision-making.

Due to the unpredictability of birth, it remains unfeasible to ensure that someone meeting full proficiency criteria [4,24]. or advanced training [5] attends every breech birth. In our earlier implementation feasibility study, 35/39 (89.7%) of births were attended by someone who

had completed full OptiBreech training [14]. In this study, this reduced to 4/7 (57.1%), for an average of 39/46 (84.8%). This may improve as services continue to embed OptiBreech team care. However, women should be informed that this cannot be guaranteed, and outcomes should be monitored.

Although the OptiBreech care pathway would be best evaluated with cluster-level randomisation, the randomisation procedures used in this pilot trial appeared feasible for use in trials to evaluate different interventions within the cohort with 1:1 randomisation. The mean gestation at recruitment was almost equivalent between arms and ideal for studies concerning management of breech presentation at term, at 35+6 and 35+5 in the randomised arms. Parity and type of breech presentation were also nearly equivalent. Other trials within the cohort, based on different questions, may be useful to refine the pathway to maximum clinical efficacy and economic efficiency. For example, complementary methods of encouraging babies to turn head-down remain popular with women and midwives, despite minimal evidence. OptiBreech sites each have dedicated clinics and referral pathways; the cohort study has tested consent procedures, randomisation methods and minimisation factors; and the ability to collect a complete dataset for key endpoints. Each of these create conditions for evaluating the effectiveness of these popular complementary therapies with maximum efficiency.

The results of the OptiBreech feasibility work also suggest that a nested randomised trial of ECV versus non-ECV for women planning a vaginal birth could offer valuable information. The Cochrane Review on *External cephalic version for breech presentation at term* indicates that attempted ECV versus non-ECV reduces the risk of caesarean birth (RR 0.57, 0.4 to 0.82) [33]. This outcome is the reason ECV is currently recommended first-line management of breech presentation at term [7]. However, it is unclear whether this effect is due to risk inherent in VBB itself or due to unwillingness to attempt a VBB, from clinicians or women themselves. Pilot trial subgroup analyses confirmed a higher risk of caesarean birth when the fetus was breech compared to cephalic on admission for labour/birth care (88.6% vs 42.1%). However, in pregnancies where a VBB was planned versus those where no VBB was planned, there was little difference in caesarean birth rate (64.7% versus 66.7%). The context of support for women who wish to attempt a VBB may have a significant effect on overall caesarean birth rates.

Current NICE antenatal guidelines also regard achieving cephalic presentation in labour as a critical outcome, [7] and current evidence suggests that ECV over non-ECV improves the chances of cephalic presentation in labour (RR 1.83, 1.53 to 2.18) [34]. However, both Cochrane and NICE report no difference in neonatal outcomes (neonatal admission and Apgar <7 at 5 minutes) between ECV versus no-ECV [33,34]. In the pilot trial's subgroup analyses, neonatal outcomes were marginally better following pregnancies where a VBB had been planned at any point compared to no VBB planned, following actual VBBs compared to cephalic vaginal births, and following admission for labour/birth care in breech presentation compared to cephalic. If confirmed in a substantive trial, this would challenge the current focus on achieving cephalic presentation in labour [7,35], rather than achieving skilled care for breech-presenting babies.

In our PPIE work and qualitative interviews with participants [1] women have raised concerns that the difficulty they experienced when trying to access support for a planned VBB would create inequities. Within our implementation feasibility study [14] and this pilot trial, over a quarter of all women needed to transfer care to another hospital, and a small number even temporarily relocated their residence, to access support for a VBB. This is only possible for women with sufficient resources and support.

Our subgroup equity analysis confirmed that the demographic participating in the OptiBreech trial is reflective of the multi-ethnic population in the UK. A marginally higher

proportion of non-British, Black or Brown women chose to plan a VBB; this only occurred within the OptiBreech care arms of the study. The increased access provided by the OptiBreech pathway may particularly benefit groups that traditionally experience increased barriers to accessing health care. In this pilot study, these groups also experienced fewer adverse neonatal outcomes. We feel this is because the continuity provided within the OptiBreech collaborative care pathway provides relational continuity and person-centred care suited to participants' individualised needs, in particular their need for breech expertise.

## Strengths and Limitations

During the 2021–2022 period in England, Hospital Episode Statistics recorded 457 spontaneous breech births and 38 instrumental breech extractions occurring after the 37th week of gestation. The implementation feasibility study [14] and this pilot trial included a total of 46 VBBs occurring in England in the same period, representing 9.3% of the total sample of term breech births nationally (46/495). Together, the feasibility studies have reported the results of 99 planned VBBs within an OptiBreech collaborative care pathway, with only 1 case of serious neonatal morbidity (1.01%). This compares well with outcomes from the Term Breech Trial, which reported a 5.5% rate of serious neonatal morbidity or mortality for planned VBB in countries with a low perinatal mortality rate, using a similar composite outcome [36].

This pilot trial is the first to compare an alternative care pathway with the current standard care pathway for management of breech birth at term. The design tests the effectiveness of the OptiBreech collaborative care model, rather than the efficacy of a specific intervention, such as ECV or planned CB. The pilot was designed to evaluate the feasibility and potential value of a substantive study, rather than detect a difference in clinical outcomes. Although it will not be possible to blind participants participating in a substantive trial, due to the similarity of mode of birth and adverse outcome rates, it should be possible to blind statisticians conducting the analysis to reduce the risk of bias.

The methods used in OptiBreech training [5,37] and delivery of the care pathway [1] were developed using rigorous, evidence-based methods and prior evaluations [1–5,38]. They have been described clearly enough in the trial protocol and developmental research to enable replication if an improvement in outcomes for VBBs is confirmed in a future trial.

## Conclusion

OptiBreech collaborative care appears to provide the person-centred care and choices expected by women and recommended in guidelines. Early results indicate it may also improve outcomes for women and babies. However, a full evaluation will require a cluster trial in sites willing to provide collaborative care.

## Supporting information

**S1 Checklist. CONSORT/GRIPP2 checklist.** Combined CONSORT checklist extension for randomised pilot and feasibility trials and GRIPP2 short form for reporting patient and public involvement in health and social care research.
(DOCX)

**S1 Fig.**
(JPEG)

**S1 File.**
(DOCX)

## Acknowledgments

Shawn Walker is funded by a National Institute of Health and Care Research (NIHR) Advanced Fellowship (300582). Jane Sandall at King's College London is an NIHR Senior Investigator. Both are supported by the NIHR Applied Research Collaboration South London (NIHR ARC South London) at King's College Hospital NHS Foundation Trust. The views expressed are those of the author(s) and not necessarily those of the NIHR or the Department of Health and Social Care, who played no role in conducting the research and writing the paper. Breech Birth Network provided training, educational resources, and funding for conference presentations.

The OptiBreech Collaborative for the pilot trial included: Avni Batish (Surrey and Sussex Healthcare NHS Trust, East Surrey Hospital), Louise Page (Chelsea and Westminster Hospital NHS Foundation Trust, West Middlesex Hospital) and Florence Wilcock (Kingston Hospital NHS Foundation Trust).

## Author Contributions

**Conceptualization:** Shawn Walker, Jane Sandall, Andrew Shennan.

**Data curation:** Shawn Walker, Emma Spillane, Kate Stringer, Lauren Trepte, Siân M. Davies, Jacana Bresson.

**Formal analysis:** Shawn Walker, Siân M. Davies.

**Funding acquisition:** Shawn Walker.

**Investigation:** Shawn Walker, Emma Spillane, Kate Stringer, Lauren Trepte.

**Methodology:** Shawn Walker, Jane Sandall, Andrew Shennan.

**Project administration:** Shawn Walker, Emma Spillane, Kate Stringer, Lauren Trepte, Siân M. Davies, Jacana Bresson.

**Supervision:** Andrew Shennan.

**Validation:** Siân M. Davies, Jacana Bresson.

**Writing – original draft:** Shawn Walker.

**Writing – review & editing:** Emma Spillane, Kate Stringer, Lauren Trepte, Siân M. Davies, Jacana Bresson, Jane Sandall, Andrew Shennan.

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
