## [Decision Letter · Decision Letter 0]

6 Jun 2023

PONE-D-23-09332OptiBreech collaborative care versus standard care for women with a breech-presenting fetus at term: a pilot parallel group randomised trial to evaluate the feasibility of a randomised trial nested within a cohortPLOS ONE

Dear Dr. Walker,

Thank you for submitting your manuscript to PLOS ONE. After careful consideration, we feel that it has merit but does not fully meet PLOS ONE’s publication criteria as it currently stands. Therefore, we invite you to submit a revised version of the manuscript that addresses the points raised during the review process.

We look forward to receiving your revised manuscript.

Kind regards,

Salvatore Andrea Mastrolia, M.D.

Academic Editor

PLOS ONE

Journal Requirements:

"I have read the journal's policy and the authors of this manuscript have the following competing interests: SW and ES are co-Directors of Breech Birth Network, Community Interest Company, a not-for-profit social enterprise that delivers VBB training and supports research. They and other members of the OptiBreech Collaborative have received teaching fees and expenses for providing breech training."

3. We note that you have referenced (Walker S, Spillane E, Stringer K, Meadowcroft A, Dasgupta T, Davies S, et al. The feasibility of team care for women seeking to plan a vaginal breech birth (OptiBreech 1) – an observational implementation feasibility study in preparation for a pilot trial. BMC Pilot & Feasibility Studies. 2022;In Review.) which has currently not yet been accepted for publication. Please remove this from your References and amend this to state in the body of your manuscript: (ie “Bewick et al. [Unpublished]”) as detailed online in our guide for authors

4. We note that the original protocol that you have uploaded as a Supporting Information file contains an institutional logo. As this logo is likely copyrighted, we ask that you please remove it from this file and upload an updated version upon resubmission.

Additional Editor Comments:

Dear Authors,

the Reviewers are favorable to the publication of your manuscript after a minor revision.

Please incorporate or discuss the suggested changes and submit a revised version of your manuscript in order to achieve publication.

Best regards

Salvatore Andrea Mastrolia

Academic Editor

Reviewers' comments:

Reviewer's Responses to Questions

**Comments to the Author**

1. Is the manuscript technically sound, and do the data support the conclusions?

Reviewer #1: Partly

2. Has the statistical analysis been performed appropriately and rigorously? 

Reviewer #1: No

3. Have the authors made all data underlying the findings in their manuscript fully available?

Reviewer #1: Yes

4. Is the manuscript presented in an intelligible fashion and written in standard English?

Reviewer #1: Yes

5. Review Comments to the Author

Reviewer #1: The primary objectives of this pilot feasibility clinical trial were to evaluate recruitment, acceptance and attrition rates for women randomized to a ‘new care process’. A statistically significantly higher proportion of women planned a VBB when randomized to OptiBreach Care. The trial terminated prematurely.

Minor revisions:

1- A. Line 239: Indicate the statistical method used for the “two-sided test.”

B. Line 295: Indicate the statistical methods used for comparing proportions. For example, 0/33 compared to 8/24 with a p = 0.003. Also remove the less than sign from the p-value.

2- Table 4: State the type of summary statistic provided following the + signs.

3- Lie 296: Indicate if the Agresti-Caffo CI is a 95% CI.

4- Line 327: State the comparison group for women randomized to OptiBreech.

5- Table 7: Define the abbreviation SD at its first occurrence.

6- Line 403: Clarify that the CI is associated with 23.5%.

6. PLOS authors have the option to publish the peer review history of their article (what does this mean?). If published, this will include your full peer review and any attached files.

Reviewer #1: No

---

## [Author Response · Author response to Decision Letter 0]

10 Jun 2023

Response to Reviewers

Checked and formatted per template.

"I have read the journal's policy and the authors of this manuscript have the following competing interests: SW and ES are co-Directors of Breech Birth Network, Community Interest Company, a not-for-profit social enterprise that delivers VBB training and supports research. They and other members of the OptiBreech Collaborative have received teaching fees and expenses for providing breech training."

We have added this statement in our cover letter. There are no restrictions on data sharing.

3. We note that you have referenced (Walker S, Spillane E, Stringer K, Meadowcroft A, Dasgupta T, Davies S, et al. The feasibility of team care for women seeking to plan a vaginal breech birth (OptiBreech 1) – an observational implementation feasibility study in preparation for a pilot trial. BMC Pilot & Feasibility Studies. 2022;In Review.) which has currently not yet been accepted for publication. Please remove this from your References and amend this to state in the body of your manuscript: (ie “Bewick et al. [Unpublished]”) as detailed online in our guide for authors

We are pleased that this paper has now been published and have updated the reference accordingly.

4. We note that the original protocol that you have uploaded as a Supporting Information file contains an institutional logo. As this logo is likely copyrighted, we ask that you please remove it from this file and upload an updated version upon resubmission.

Changed and version without logos uploaded.

Done.

This has been checked. The only changes are formatting and the full reference for the paper recently published. To the best of our knowledge, we have not cited any retracted articles.

Additional Editor Comments:

Dear Authors,

the Reviewers are favorable to the publication of your manuscript after a minor revision.

Please incorporate or discuss the suggested changes and submit a revised version of your manuscript in order to achieve publication.

Best regards

Salvatore Andrea Mastrolia

Academic Editor

Wonderful news! Thank you for your team’s careful review and guidance to improve this manuscript.

Reviewers' comments:

Review Comments to the Author

Reviewer #1: The primary objectives of this pilot feasibility clinical trial were to evaluate recruitment, acceptance and attrition rates for women randomized to a ‘new care process’. A statistically significantly higher proportion of women planned a VBB when randomized to OptiBreach Care. The trial terminated prematurely.

Minor revisions:

1- A. Line 239: Indicate the statistical method used for the “two-sided test.”

Thank you. We have now clarified that a chi-square test was used.

B. Line 295: Indicate the statistical methods used for comparing proportions. For example, 0/33 compared to 8/24 with a p = 0.003.

Thank you. This has been clarified with a detailed description. The denominator in the second proportion is 34 (consistent with original manuscript submission).

Also remove the less than sign from the p-value.

Removed.

2- Table 4: State the type of summary statistic provided following the + signs.

We have clarified in the table that this refers to weeks + days gestation.

3- Lie 296: Indicate if the Agresti-Caffo CI is a 95% CI.

95% added.

4- Line 327: State the comparison group for women randomized to OptiBreech.

Stated.

5- Table 7: Define the abbreviation SD at its first occurrence.

Defined in Table 6.

6- Line 403: Clarify that the CI is associated with 23.5%.

Done.

We hope this has satisfactorily addressed the statistical reviewer’s concerns. Many thanks again for your careful attention to detail, which has improved this manuscript.

---

## [Decision Letter · Decision Letter 1]

18 Oct 2023

PONE-D-23-09332R1OptiBreech collaborative care versus standard care for women with a breech-presenting fetus at term: a pilot parallel group randomised trial to evaluate the feasibility of a randomised trial nested within a cohortPLOS ONE

Dear Dr. Walker,

Thank you for submitting your manuscript to PLOS ONE. After careful consideration, we feel that it has merit but does not fully meet PLOS ONE’s publication criteria as it currently stands. Therefore, we invite you to submit a revised version of the manuscript that addresses the points raised during the review process.

We look forward to receiving your revised manuscript.

Kind regards,

Astawus Alemayehu

Academic Editor

PLOS ONE

Journal Requirements:

Additional Editor Comments:

I would like to suggest the author to address reviewer 1 comment "In the reply the author indicate that proportions were compared with the chi-square test. In addition, the chi-square test should be noted in the manuscript."

Reviewers' comments:

Reviewer's Responses to Questions

**Comments to the Author**

1. If the authors have adequately addressed your comments raised in a previous round of review and you feel that this manuscript is now acceptable for publication, you may indicate that here to bypass the “Comments to the Author” section, enter your conflict of interest statement in the “Confidential to Editor” section, and submit your "Accept" recommendation.

Reviewer #1: (No Response)

Reviewer #2: All comments have been addressed

Reviewer #3: All comments have been addressed

2. Is the manuscript technically sound, and do the data support the conclusions?

Reviewer #1: Yes

Reviewer #2: Yes

Reviewer #3: Yes

3. Has the statistical analysis been performed appropriately and rigorously? 

Reviewer #1: Yes

Reviewer #2: Yes

Reviewer #3: Yes

4. Have the authors made all data underlying the findings in their manuscript fully available?

Reviewer #1: Yes

Reviewer #2: Yes

Reviewer #3: Yes

5. Is the manuscript presented in an intelligible fashion and written in standard English?

Reviewer #1: Yes

Reviewer #2: Yes

Reviewer #3: Yes

6. Review Comments to the Author

Reviewer #1: Minor revision:

In the reply the author indicate that proportions were compared with the chi-square test. In addition, the chi-square test should be noted in the manuscript.

Reviewer #2: I find this project to be truly intriguing, and I would like to express my appreciation to the authors for diligently addressing the previous comments. I recommend publishing this paper as it significantly contributes to the existing literature. However, I do have a question regarding the excessive use of the authors' own references, which appears to exceed 16 instances. I wonder if this is a deliberate strategy to increase citations to their previous work. Nevertheless, despite this concern, I maintain a favorable opinion of the paper and believe it offers valuable insights to the field.

Reviewer #3: I would like to thank and congratulate the authors for their interesting work, now the manuscript is fit or suitable for publication.

7. PLOS authors have the option to publish the peer review history of their article (what does this mean?). If published, this will include your full peer review and any attached files.

Reviewer #1: No

Reviewer #2: **Yes: **Mena Abdalla

Reviewer #3: **Yes: **Abebaw Demissie Woldemariam

---

## [Author Response · Author response to Decision Letter 1]

18 Oct 2023

Additional Editor Comments:

I would like to suggest the author to address reviewer 1 comment "In the reply the author indicate that proportions were compared with the chi-square test. In addition, the chi-square test should be noted in the manuscript."

It is more correct to say we used a Fisher exact test, as one of the cells contained zero. This is now noted in the manuscript (line 234).

Reviewer's Responses to Questions

Comments to the Author

Reviewer #1: Minor revision:

In the reply the author indicate that proportions were compared with the chi-square test. In addition, the chi-square test should be noted in the manuscript.

It is more correct to say we used a Fisher exact test, as one of the cells contained zero. This is now noted in the manuscript (line 234).

Reviewer #2: I find this project to be truly intriguing, and I would like to express my appreciation to the authors for diligently addressing the previous comments. I recommend publishing this paper as it significantly contributes to the existing literature. However, I do have a question regarding the excessive use of the authors' own references, which appears to exceed 16 instances. I wonder if this is a deliberate strategy to increase citations to their previous work. Nevertheless, despite this concern, I maintain a favorable opinion of the paper and believe it offers valuable insights to the field.

This is a reasonable concern! However, the strategy is to demonstrate that this complex intervention is the result of over a decade’s worth of theoretical work and evaluation of its components. We have also become aware of some guidelines using our work, particularly the time interval recommendations in the Physiological Breech Birth Algorithm, without citation. We want to be clear about how the intervention has been developed and that we are not just testing ‘vaginal breech birth’ according to the individual provider’s preferences.

Two of the citations are for the protocol registration and the data repository. K Walker is no relation and not part of the current research team.

Nonetheless, we will take this to heart and write an article that dissects the complex intervention. Then we will be able to refer succinctly to one summary publication in the future.

Reviewer #3: I would like to thank and congratulate the authors for their interesting work, now the manuscript is fit or suitable for publication.

Thank you.

---

## [Editor Report · Decision Letter 2]

26 Oct 2023

OptiBreech collaborative care versus standard care for women with a breech-presenting fetus at term: a pilot parallel group randomised trial to evaluate the feasibility of a randomised trial nested within a cohort

PONE-D-23-09332R2

Dear Dr. Walker,

We’re pleased to inform you that your manuscript has been judged scientifically suitable for publication and will be formally accepted for publication once it meets all outstanding technical requirements.

Kind regards,

Astawus Alemayehu Feleke, Ph.D Can..

Academic Editor

PLOS ONE

Additional Editor Comments (optional):

I would like to congratulate the authors for their valuable manuscript which may contribute clue for the field, Programmer and health policy.
---

## [Editor Report · Acceptance letter]

6 Nov 2023

PONE-D-23-09332R2 

OptiBreech collaborative care versus standard care for women with a breech-presenting fetus at term: a pilot parallel group randomised trial to evaluate the feasibility of a randomised trial nested within a cohort 

Dear Dr. Walker:

I'm pleased to inform you that your manuscript has been deemed suitable for publication in PLOS ONE. Congratulations! Your manuscript is now with our production department. 

Kind regards, 

on behalf of

Dr. Astawus Alemayehu Feleke 

Academic Editor

PLOS ONE